# Gendered Working Environments as a Determinant of Mental Health Inequalities: A Protocol for a Systematic Review

**DOI:** 10.3390/ijerph16071169

**Published:** 2019-04-01

**Authors:** Allison Milner, Anna J. Scovelle, Tania L. King, Claudia H. Marck, Ashley McAllister, Anne M. Kavanagh, Marissa Shields, Eszter Török, Adrienne O’Neil

**Affiliations:** 1Centre for Health Equity, Melbourne School of Population and Global Health; University of Melbourne, Melbourne 3010, Australia; anna.scovelle@unimelb.edu.au (A.J.S.); tking@unimelb.edu.au (T.L.K.); claudia.marck@unimelb.edu.au (C.H.M.); a.kavanagh@unimelb.edu.au (A.M.K.); marissa.shields@unimelb.edu.au (M.S.); 2Equity and Health Policy, Department of Public Health Sciences; Karolinska Institute, 162 54 Stockholm, Sweden; ashley.mcallister@ki.se; 3Department of Public Health; University of Copenhagen, 2300 Copenhagen, Denmark; eszt@sund.ku.dk; 4Centre for Mental Health, School of Population and Global Health; University of Melbourne, Melbourne 3010, Australia; adrienne.oneil@unimelb.edu.au

**Keywords:** gender, employment, work, mental health, inequalities, review

## Abstract

Both gender and employment are critical and intersecting social determinants of mental and physical health. This paper describes the protocol used to conduct a systematic literature review of the relationship between “gendered working environments” and mental health. Gendered working environments (GWE) are conceptualised as involving: (1) differences in selection into work, and more specifically, occupations; (2) variation in employment arrangements and working hours; (3) disparities in psychosocial exposures at work, and; (4) differences in selection out of work. Methods/design: The review will adhere to a Preferred Reporting Items for Systematic Reviews and Meta-Analyses (PRISMA) search procedure. Key words will be identified that are specific to each of the four domains of GWE. The databases used for the search will be Scopus, Pubmed, Proquest, and Web of Science. Keywords will be adapted for the specific requirements of each electronic database. Inclusion criteria are: Using a validated scale to measure mental health (outcome); including exposures related to the four domains of GWE; reporting estimates for both men and women; and use of a cohort, case-control, or cross-sectional design. Studies will be excluded if they were published more than 10 years ago, are not in English or do not present extractable data on the relationship between GWE and mental health. Discussion: The proposed review will provide evidence about the numerous and complex ways in which employment and gender intersect (and are reinforced) to influence mental health over the life course.

## 1. Introduction

There is a long history of research demonstrating gender differences in employment. For example, in high-income countries, men are much more likely to work full-time than women [1]. There are also gender differences in the type of occupations in which men and women are employed, with a greater proportion of women employed in health and human service work [2]. Men, on the other hand, are more likely to be employed in management roles, and in construction and manufacturing [2]. Women are more likely to have “interrupted” working careers due to caring responsibilities, and thus may have a working life characterised by entry and exit from the labour market. Data have shown that women have a notable “wage penalty” per child, compared to men whose wage increases as a result of having children (“motherhood penalty” versus “the fatherhood bonus” [3]). These gendered labour force patterns intersect with other key determinants of employment and health, such as ethnicity, age, and socio-economic status [4].

An understanding of the gendered causes of employment inequalities is highly relevant to public health research and policy, given the recognition that work is an important social determinant of health [5]. There is a substantial amount of research demonstrating that exposure to unemployment affects physical and mental health [6], as well as being associated with higher mortality [7]. There is also considerable evidence to show that people working in jobs with exposure to psychosocial job stressors (e.g., low control over when, where and how work is undertaken, insecurity, poor supervisor or collegial support) have worse mental health outcomes than those persons working in jobs with better working conditions [8,9,10]. We argue that gender is critical in understanding these employment-related health inequalities.

## 2. Conceptualisation of Gendered Working Environments

### 2.1. Definition of Gender as a Structural Determinant of Health

Gender and sex are related, but separate constructs. Gender refers to a social construct regarding culture-bound conventions, roles, and behaviours, as well as relations between and among, women and men and boys and girls [11]. Gender relations vary within and across societies, typically in relation to social divisions premised on power and authority (e.g., class, race/ethnicity, nationality, religion). In comparison, sex is a biological construct premised upon biological characteristics [11].

### 2.2. Definitions of Gendered Working Environments

There are of course a multitude of perspectives on how the relationship between gender and employment have evolved and continue to evolve over time, but we are particularly interested in the issues of differences in exposure to various aspects of work, such as the type of work undertaken, working arrangements, psychosocial exposures while at work, and differences in how women and men exit from work. It is perhaps most simple to imagine this concept from a life course perspective, starting when young people enter employment and continue throughout their working life. For example, women may be selected into more female dominated occupations, which determines their exposure to certain working arrangements, pay and psychosocial working conditions. Employment in these jobs may contribute to different reasons for leaving the workforce, for either unemployment or other reasons such as to care for dependents. It is also necessary to highlight the fact that there may be differences in physical and biological exposures experienced by men and women, although in this in this article, we are particularly interested in psychosocial exposures.

Based on this, we have conceptualised at least four domains of gendered working environments (Figure 1):Differences in selection into work, and more specifically, into occupations.Variation in employment arrangements and working hours, including “flexible” or precarious employment, casual or part-time work, underemployment and (the converse) long working hours, and temporary absence from work due to sick leave, parental leave, etc.Differences in psychosocial exposures at work.Differential in selection out of work.

We explain each of these domains in greater depth below and in Figure 1.

### 2.3. Differential Selection into Work

From an early age, children and adolescents show clear preferences in terms of their career aspirations [12,13,14]. For example, a large cohort study in the United Kingdom [14] demonstrated that, at age seven, a large proportion of boys and girls show clear differences in their aspirations of work. This is likely to be driven by social conditioning about the roles and norms that are seen as appropriate for women and men. A study by Howard et al. [12] across 22,000 young people in the United States and noted that these gender differences in aspiration persisted into adolescence. These studies are important because of the link between the career aspirations of young people and adult career attainment [13].

On reaching working age, the majority of employed males enter into work within male-dominated occupations such as police, trades work, information technology and machinery-based work. The majority of females work in education, health, hospitality, sales work, or clerical and administrative positions [2]. As we discuss below, this initial gendered divide in occupations has implications for subsequent psychosocial exposures.

### 2.4. Variation in Employment Arrangements and Working Hours

There are clear gender differences in working arrangements and hours worked among men and women [15]. Males generally work longer paid hours than females [1,16]. Moreover, males are more likely to be overworked (at least in terms of hours in paid employment) while females are more likely to be underemployed (where they would work more if the hours were offered to them by an employer) [17]. At the same time, women are more likely to be employed on a part-time, or temporary basis than men [18]. Precarious employment (work with short term contracts or temporary working arrangements) is more prevalent among women [19]. These variations are particularly notable in midlife, when a substantial proportion of males and females are taking parental leave and looking after young children [20,21]. It is also worth noting that, at all ages, females take on a greater proportion of home-based work [22]. This is despite increased rates of women’s education and employment over the past four decades. For example, in the US, a greater number of women than men have graduated with Bachelors, Masters and Doctorate degrees [23] since the 1980s. The topic of the “work-family interface” is not something we explicitly considered in the concept of gendered working environments, because it refers to home stressors occurring outside the workplace. However, it is pertinent to selection effects in and out of the workforce by gender (as discussed below).

### 2.5. Differences in Psychosocial Exposures

There has been a notable amount of research regarding gender differences in exposure to psychosocial job stressors such as job control, job demands, social support at work, and effort-reward imbalance, as summarised in previous reviews [19,24,25,26,27]. Research conducted since the time these reviews were published has suggested that women may have greater exposure to more common psychosocial job stressors than men [28]. For example, a recent cross sectional study from Canada [28] reported that men had higher skill utilisation and decision authority than women. Another study in Japan [29] suggested that women reported lower job control and higher effort-reward imbalance than men. A general population study in Norway [30] suggested that women were much more likely to report exposure to low job control, high demands, and low support than men. Results from France [31] suggested that women have greater exposure to a range of psychosocial job stressors compared to men, including job strain (the combination of high demands and low latitude) and iso-strain (the combination of job strain and low support). This diversity among studies highlighted the likely role of differences in sample composition and contextual differences. Aside from these stressors, there is increasing recognition of the importance of other stressors such as violence at work, which can affect both women and men, particularly in the health care profession [32,33].

### 2.6. Differential Selection Out of Work

Aside from differences in working arrangements and psychosocial exposures, there are also gender specific reasons for exiting employment (leaving work). In some Organisation for Economic Co-operation and Development (OECD) countries, women have higher rates of unemployment than men on average, while the opposite is the case in other countries [34]. There may also be notable differences in retirement [35]. For example, some studies have suggested that women are much more likely to retire earlier than men [35], while other research suggested that there was no difference in retirement between women and men [36]. Women in lower skilled occupations are more likely to leave employment than their male counterparts [37]. Macro-economic conditions appear to be important, with women being more likely to retire early when unemployment rates are higher rather than lower [35]. Retirement policies are also an important factor, with some countries having lower retirement ages for women, compared to men.

Having defined the concept of gendered working environments, we intend to describe a protocol for a systematic review about what is known regarding the mental health effects of gendered working environments, by searching the relevant published research. We chose mental health of women and men as an outcome of the review as this is particularly sensitive to changes in employment status and conditions [38,39].

## 3. Methods and Analysis

### 3.1. Search Strategy and Keywords

We will conduct this review utilizing the Preferred Reporting Items for Systematic Reviews and Meta-Analyses (PRISMA) search approach [40]. Key words will be identified specific to each of the four areas above (Appendix A). The databases used for the search will be Scopus, PubMed, Proquest, and Web of Science. Keywords will be adapted for the specific requirements of each electronic database. Truncation and wildcards will be introduced where necessary to increase the sensitivity of the search. The PRISMA-P checklist can be seen in Appendix A.

### 3.2. Inclusion and Exclusion Criteria

Studies have shown that a change or comparison in mental health status using a validated mental health measure will be considered as eligible. We are particularly interested in the measurement of the common mental disorders of depression and anxiety. We will also consider physician diagnosed cases. Aside from this, we are particularly interested in evidence from quantitative studies, including cohort, case-control, or cross-sectional designs. Studies will provide an estimate of work-related exposures for men and women, in relation to mental health outcomes. It is important to note here that many studies will conflate sex and gender, use the terms inter-changeably or treat sex/gender in a binary fashion; all common oversights in scientific research. Accordingly, the decision was taken by the Authorship Group to include papers where these nuances and distinctions were not explicitly made, given their expected high prevalence. Limitations of this approach will be explicated in the context of the Discussion section of the review paper. We are particularly interested in studies published in the last ten years as a previous review covered the period up until 2010 [19], which accompanied previous narrative reviews on the topic of gender and work [24,25,26,27]. Only articles that are published in English will be considered. Furthermore, only research published in peer-reviewed articles will be considered. We will exclude studies that are purely qualitative, reviews, or case reports. Studies measuring suicide and self-harm will be excluded. “Grey” literature will also be excluded as these may not have undertaken a peer review process.

### 3.3. Search Procedure

We will use Endnote (Clarivate Analytics, Philadelphia, PA, USA) and Microsoft Excel to download and organise the references. After relevant publications are identified and duplicates removed, titles and abstracts will be searched for keywords by at least two researchers. Publications identified for further review will undergo full-text screening by eight of the authors in order to determine their eligibility. Following full text review, data from eligible publications will be extracted by eight of the authors, working independently. The data extraction forms have been piloted by two of the authors and found to be acceptable for use.

### 3.4. Data Extraction

We will extract information on the year and country of the study, study design and the sample used in the study. We will also extract information on key employment exposure, the mental health measurement used and results of the study. We will undertake a quality review of studies, based on an adapted version of the Newcastle Ottawa Quality Assessment Scale [41]. A total quality score will be produced for each study, by summing the number of domains that meet each criterion.

### 3.5. Data Synthesis

Information regarding the outcomes and exposures (aspects of the gendered working environment) will be extracted and summarised in descriptive tables and described in the text. No formal quantitative analysis will be undertaken. Rather, we will only consider a narrative/descriptive synthesis. The quality assessment (described above) will be discussed as part of the synthesis, and poor quality studies will be identified. The potential impact of the findings of the poor quality studies on the overall synthesis will be discussed. At least two authors will be involved in the narrative synthesis. Any discrepancies between the authors regarding the synthesis will be resolved by a third author.

Ethics approval and consent to participate: Not applicable.

## 4. Discussion

This review will provide a comprehensive assessment of the ways in which gender intersects with employment throughout the life-course. This includes information about how differential selection into employment impacts on the mental health of women and men. We will also explore the mental health impacts of any variation in employment arrangements and working hours, exposure to psychosocial job stressors, and any differences about how men and women are selected for the workforce. Ultimately, this will provide a framework from which to consider the gendered impact of employment on working men and women, as well as how their experiences of work may have cumulative effects on their mental health. This is likely to be useful to researchers and policy makers across health promotion, organizational psychology and other related disciplines.

## 5. Conclusions

The proposed review will provide evidence about the numerous and complex ways in which employment and gender intersect (and are reinforced) to influence mental health over the life course. In this paper, we provide the rationale for the concept of “gendered working environments” which describes gender specific differences in selection into work (1); variation in employment arrangements and working hours (2); differences in psychosocial exposures at work (3), and differential in selection out of work (4).

## Figures and Tables

**Figure 1 ijerph-16-01169-f001:**
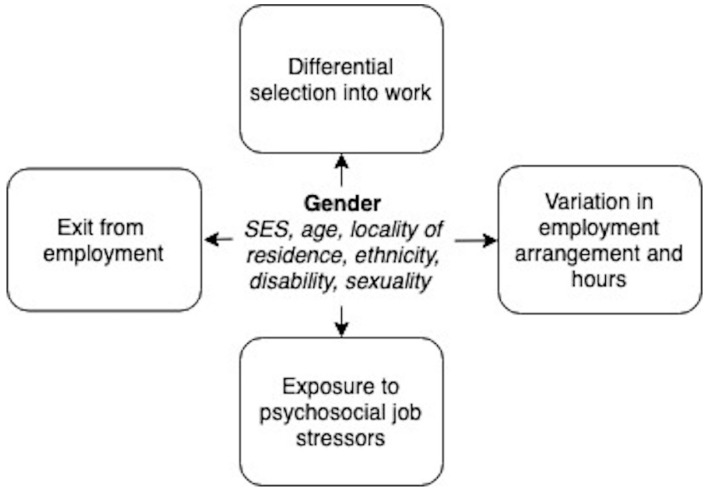
Conceptualisation of gendered working environments.

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
