# Peer review of "Gendered Working Environments as a Determinant of Mental Health Inequalities: A Protocol for a Systematic Review"

_ijerph, 2019, doi:10.3390/ijerph16071169_

Round 1

Reviewer 1 Report

The issue of gendered working environments is important and timely.

However, limiting this proposed systematic review to the last 10 years without presenting at least a summary of the previous 10 years prevents the reader from the necessary comparison, especially given the economic crisis of 2007-8 and its consequences.

The inclusion of grey literature is recommended, as the subjective and inter-subjective experience dimension will add new information not covered by the more formalized studies.

This far we learn too little about the data analysis and its dimensions.

Author Response

Comment 1:

The issue of gendered working environments is important and timely. However, limiting this proposed systematic review to the last 10 years without presenting at least a summary of the previous 10 years prevents the reader from the necessary comparison, especially given the economic crisis of 2007-8 and its consequences. The inclusion of grey literature is recommended, as the subjective and inter-subjective experience dimension will add new information not covered by the more formalized studies. This far we learn too little about the data analysis and its dimensions.

Response 1:

The reason that the scope of the review was confined to the previous ten years was because there has been previous reviews on gender an work covering the years up until 2010 (Campos-Serna et al., 2013), which accompanies other narrative reviews of the topic (e.g., Artazcoz et al., 2007; Härenstam, 2009). This has been added into the paper (page 5, para 1):

We are particularly interested in studies published in the last ten years as a previous review covered the period up until 2010 (19), which accompany previous narrative review on the topic of gender and work (24-27).

Grey literature was not included for two reasons. First, the literature on this topic is considerable, as evidenced in the previous review. Second, there is no guarantee as to the quality of grey literature, which has not usually gone through the peer review process. For these two reasons, we decided it was best not to include this literature and have added this justification into the paper (page 5, para 1):

“Grey” liteature will be excluded as this may not have passed through the peer review process.

References

Artazcoz L, Borrell C, Cortàs I, et al (2007). Occupational epidemiology and work related inequalities in health: a gender perspective for two complementary approaches to work and health research Journal of Epidemiology & Community Health;61:ii39-ii45.

Campos-Serna J, Ronda-Pérez E, Artazcoz L, Moen BE, Benavides FG (2013). Occupational epidemiology and work related inequalities in health: a gender perspective for two complementary approaches to work and health research. Int J Equity Health;12:57. doi: 10.1186/1475-9276-12-57.

Härenstam A (2009). Exploring gender, work and living conditions and health - suggestions for contextual and comprehensive approaches. Scand J Work Environ Health ;35(2):127-33

Reviewer 2 Report

Thank you for the opportunity to review this manuscript.This is an important topic and the background and introduction sets out a good rationale for the review. However, the manuscript is incomplete and although it sets out to describe a protocol for a systematic review on gendered working environments as a determinant of mental health inequalities there several elements of the protocol which are not defined. I suggest that the systematic review when completed is resubmitted. 

Author Response

Comment 1:

Thank you for the opportunity to review this manuscript. This is an important topic and the background and introduction sets out a good rationale for the review. However, the manuscript is incomplete and although it sets out to describe a protocol for a systematic review on gendered working environments as a determinant of mental health inequalities there several elements of the protocol which are not defined. I suggest that the systematic review when completed is resubmitted. 

Response 1:

We appreciate the reviewer noting the importance of this review, however we are unsure what the reviewer means by the protocol being incomplete. We have recently submitted the protocol with PROSPERO, which is an internationally accepted register of systematic reviews that requires compliance with the PRISMA-P checklist. Our protocol has been reviewed by the PROSPERO team and only one question pertaining to the data synthesis was raised. We submitted our response to PROSPERO on March 7th and expect formal approval and registration with PROSPERO shortly. We can advise of the registration number once received.

We have now included our response to PROSPERO in the paper (page 5, para 4):

No formal quantitative analysis will occur. Rather, we will consider only a narrative/descriptive synthesis. The quality assessment (described above) will be discussed as part of the synthesis, and poor quality studies will be identified. The potential impact of the findings of the poor quality studies on the overall synthesis will be discussed. At least two authors will be involved in the narrative synthesis. Any discrepancies between the authors regarding the synthesis will be resolved with a third author.”

Also, we thank the reviewer for this suggestion of submitting the completed review back to the journal. We will certainly consider this.

Round 2

Reviewer 2 Report

The corrections have added to the clarity of the manuscript. It is giood to learn that the study has been submitted to PROSPEROUS and as soon as the registration is complete the authors should submit the registration number with the manuscript.